# Plasma Membrane-Derived Liposomes Exhibit Robust Antiviral Activity against HSV-1

**DOI:** 10.3390/v14040799

**Published:** 2022-04-12

**Authors:** Ilina Bhattacharya, Tejabhiram Yadavalli, David Wu, Deepak Shukla

**Affiliations:** 1Department of Ophthalmology and Visual Sciences, University of Illinois at Chicago, Chicago, IL 60612, USA; ibhatt4@uic.edu (I.B.); yteja@uic.edu (T.Y.); dwu36@uic.edu (D.W.); 2Department of Microbiology and Immunology, University of Illinois at Chicago, Chicago, IL 60612, USA

**Keywords:** plasma membrane, antiviral, herpesviruses, virus neutralization, therapy

## Abstract

Plasma membranes host a plethora of proteins and glycans on their outer surface that are exploited by viruses to enter the cells. In this study, we have utilized this property to limit a viral infection using plasma membrane-derived vesicles. We show that plasma membrane-derived liposomes are prophylactically and therapeutically competent at preventing herpes simplex virus type-1 (HSV-1) infection. Plasma membrane liposomes derived from human corneal epithelial (HCE) cells, which are natural targets of HSV-1 infection, as well as Vero and Chinese hamster ovary (CHO) cells were used in this study. Our study clearly demonstrates that HCE and Vero-derived cellular liposomes, which express the viral entry-specific cell surface protein receptors, exhibit robust antiviral activity especially when compared to CHO-derived liposomes, which lack the relevant HSV-1 entry receptors. Further experimentation of the plasma membrane-derived liposomes with HSV type-2 (HSV-2) and pseudorabies virus yielded similar results, indicating strong potential for the employment of these liposomes to study viral entry mechanisms in a cell free-environment.

## 1. Introduction

Plasma membranes form the outermost layer of mammalian cells, made up of lipids that host a variety of glycans, proteins and proteoglycans within their bilayer structure [1]. Being the outermost protective layer, plasma membranes are the primary barrier for any viral infections [2]. Plasma membranes consist of various protein receptors that viruses have evolved to use to their own advantage to attach, fuse and enter cells [2,3]. A host cell contains various membrane proteins that act as cellular entry receptors for multiple different viruses. For example, to initiate HSV-1 entry into the host cells; viral glycoproteins gB, gD, gH and gL are required to interact with herpesvirus entry mediator (HVEM), nectins 1 and 2, and 3-*O*-sulfated heparan sulfate and coronaviruses use human angiotensin-converting enzyme-2 (ACE2) receptor for the same purpose [4,5]. HVEMs are usually expressed on human T cells [4], whereas nectins mediate entry of HSV-1 into human epithelial and neuronal cells [4].

Other research studies have shown that purified proteins of entry receptors can be used as natural targets to neutralize viruses, efficiently blocking them from entering cells [6]. In this regard, we hypothesized that particles that contain these receptor proteins for various viruses can be used as broad-spectrum neutralizing tools against them. Given that the plethora of receptor proteins required by virus are already innately entrenched in a lipid-based plasma membrane, it would be ideal to prepare neutralizing particles with them.

Particles of lipid origin, such as liposomes, are very commonly used as drug carriers [7,8,9,10]. These liposomes are biocompatible, non-toxic and can be modified based on the necessity of the application. Plasma membranes, as mentioned earlier, are made up of lipids. If processed correctly, they can be converted into liposome-like structures [11,12,13,14,15]. In this study, we used previously validated techniques and commercially available kits to isolate cellular plasma membranes and convert them into plasma membrane-based neutralizing particles (PMNPs). We used HSV-1 permissive (Vero and HCE) and non-permissive (CHO) cell lines to create our PMNPs. The non permissive cell lines lack HSV-1 entry receptors HVEM and nectin and thus cannot facilitate HSV-1 entry [16]. These particles were then used to neutralize multiple herpesviruses including herpes simplex virus type-1 (HSV-1).

## 2. Materials and Methods

### 2.1. Cells, Viruses, Media, Buffers and Antibodies

Human corneal epithelial (HCE), Chinese hamster ovary (CHO), and Vero cell lines were used in experimentation. The HCE cell line was obtained from Kozaburo Hayashi (National Eye Institute, Bethesda, MD, USA) [17] and cultured in MEM (Gibco, Carlsbad, CA, USA) with 10% fetal bovine serum (FBS) and 1% penicillin/streptomycin (P/S) (Sigma-Aldrich, St. Louis, MO, USA). CHO cells were obtained from P. G. Spear (Northwestern University) and passaged in F12 medium (Gibco, Carlsbad, CA, USA) with 10% FBS and 1% P/S. Vero cells, derived from African green monkey kidney cells, were obtained from the American Type Culture Collection (Manassas, VA, USA) and cultured in Dulbecco’s modified Eagle’s medium (DMEM, Gibco, Carlsbad, CA, USA) supplemented with 10% FBS and 1% P/S. HSV-1 17syn+ and HSV-1 17gfp (HSV-1 17syn+ modified with an insertion of a GFP reporter gene) strain [18] were provided by Dr. Richard Thompson (University of Cincinnati, Cincinnati, OH, USA). Red fluorescent protein (RFP)-HSV-1 (K26-GFP) with KOS strain background were provided by Prashant Desai (Johns Hopkins University, Maryland, MD, USA). All the viruses were made and titered on Vero cells and stored at −80 °C until use.

The following antibodies were used for western blot:

Mouse monoclonal anti-GAPDH (Santa Cruz, sc-69778, [7B]) (1:1000), Mouse monoclonal anti-HSV1 gD (Abcam, Cambridge, UK, ab18638, [1-I-9]) (1:1000) Goat anti-Mouse IgG (H + L) Highly Cross-Adsorbed Secondary Antibody HRP (Thermo Fisher, Waltham, MA, USA, 31432) (1:5000), Goat anti-Rabbit IgG (H  +  L) Cross Adsorbed Secondary Antibody HRP (Thermo Fisher, G-21234) (1:5000), Mouse monoclonal Anti-Nectin 1 Antibody (F-10)(Santa Cruz, sc-271063) (1:1000), Rabbit monoclonal Anti-Sodium Potassium ATPase antibody (Abcam, ab58475) (1:1000).

### 2.2. Plasma Membrane Isolation

#### 2.2.1. Manual Isolation

Two different methodologies were employed for the isolation of plasma membranes from cell cultures. In the first manual method, which was adapted from an earlier report [19], HCE and CHO cells were lysed in radioimmunoprecipitation assay (RIPA) buffer (Sigma, Cat no: 20-188) and DI water using three repeated freeze–thaw cycles. The lysed samples were vortexed and sonicated in a Fisher Scientific FB120 Sonic Dismembrator at 20 kHz frequency for 30 s on ice. Since this lysis was quite vigorous, the proteins were protected with a protease and phosphatase inhibitor. Following cell lysis, the cell nuclei were separated from the remaining cellular components with two centrifugations at 800× *g*; the supernatant was preserved, and the pellet was discarded. The mitochondria were then separated in the same way with two centrifugations at 10,000× *g*. Finally, the plasma membrane was separated from the cellular proteins with two successive centrifugations at 25,000× *g*. The supernatant was discarded, and the plasma membranes remained in the pellet. The plasma membrane pellet was resuspended in 1 mL of fresh PBS and sonicated as described above to form the PMNPs.

#### 2.2.2. Plasma Membrane Isolation Kit

Alternatively, a multi-stage centrifugation kit was employed to isolate the membranes (Invent Biotechnologies Minute Plasma Membrane Protein Isolation and Cell Fractionation Kit). Cell isolates were passed through a filter tube via centrifugation after first being washed with buffer A. The cell isolates were then washed with buffer B and centrifuged again. Repeated centrifugations were used to isolate the final product from the nuclei and the cytosol, and finally, the pellets containing the plasma membrane liposomes were stored at −80 °C until use. The plasma membrane pellet was resuspended in 1 mL of fresh PBS and sonicated as described to form the PMNPs.

### 2.3. Neutralization Assay

HCE plasma membrane-derived liposomes and their CHO counterparts were combined with HSV-1 17 virus in 1.5 mL Eppendorf tubes and agitated to promote mixing. After 30 min of incubation at room temperature, the contents were centrifuged at 10,000× *g* and plated onto Vero cells.

### 2.4. Viral Entry Assay

The viral entry assay was performed using protocols previously described by our group [20]. Briefly, HCEs were seeded in a 96-well plate 16 h prior to the start of the experiment to ensure approximately 10,000 cells per well. Plasma membrane stocks were diluted at different concentrations in OptiMEM and mixed with 100,000 PFU of reporter virus HSV-1 gL86 per well. A portion of the “gL” gene was substituted with lac Z gene encoding for the β-galactosidase enzyme in HSV-1 gL86 virus [21]. This solution mixture, containing different concentrations of the plasma membrane and gL86, was incubated for a period of 30 min at room temperature prior to adding them to HCEs. At 6 h after the addition of the mixture, cells were washed with PBS and 100 µL/well of staining mixture (0.1% Nonidet P40 and 36 mg of ortho-Nitrophenyl-β-galactoside in 12 mL of PBS) was added. After 30 min of incubation, colorimetric differences between the samples were analyzed using a plate reader at 405 nm.

### 2.5. Transmission Electron Microscopy Imaging

Sonicated PMNPs were fixed in 1% Glut + 4% p-FA, (pH 7.2), post-fixed with 1% osmium tetroxide (1 h), and dehydrated using an ascending series of ethanol (through 100% absolute). They were then embedded in LX112 epoxy resin and polymerized at 60 °C for 3 days. Thin sections (~75 nm) were collected onto copper grids and stained with uranyl acetate and then lead citrate. Specimens were examined using a JEOL JEM-1400F transmission electron microscope (at 80 kV). Micrographs were acquired using an AMT Side-Mount Nano Sprint Model 1200S-B and Biosprint 12M-B cameras, loaded with AMT Imaging software V.7.0.1.

### 2.6. Cell Viability Assay

The cell viability in presence of PMNPs was assessed by performing the MTT assay. HCE cells were seeded at a density of 4 × 10^4^/well in a 96-well plate. Confluent cells were subjected to indicated concentrations of PMNPs. Following 24 h incubation, cells were washed with DPBS, and then 10 µL of MTT reagent (5 mg/mL) was added to each well. After 4–6 h of incubation, the formation of purple formazan crystals was observed. A total of 100 µL of acidified isopropanol (1% glacial acetic acid *v*/*v*) was added to dissolve the formazan crystals. After complete solubilization of the purple formazan crystals, the spectrophotometric absorbance of the formazan product was measured by a microplate reader at 560 nm.

### 2.7. Flow Cytometry

HSV-1 17GFP-infected samples were collected at 24 hpi using Hank’s based enzyme-free cell dissociation buffer (Gibco) and centrifuged for 5 min at 800 RCF. Cell pellets were washed with PBS and resuspended in 1% FBS in PBS. The flow cytometry was conducted using a BD Accuri C6 Plus machine to detect GFP positive cells to indicate a productive infection. A total of 10,000 events were collected for each sample within the chosen gate, with analysis performed with FlowJo (version 10).

### 2.8. Western Blot

Samples were collected in Hank’s based enzyme-free cell dissociation buffer (Gibco) and centrifuged for 10 min at 800× *g*. The supernatant was then removed from the samples, and 50 µL of radioimmunoprecipitation assay buffer was added to each cell pellet. After 30 min of incubation on ice, the samples were centrifuged for 30 min at 12,000× *g* at 4 °C followed by mixing the supernatants with β-mercaptoethanol and LDS buffer and then were loaded into precast SDS-PAGE gels. Electrophoresis was performed and the protein lanes were transferred to a nitrocellulose membrane using a fast-dry transfer instrument (Invitrogen iBlot2). Membranes were blocked in 5% skimmed milk in TBS-T (Tris-buffered saline with 1% Tween-20) for 1 h.

### 2.9. Plaque Assay

Infected treated and infected non-treated cells were collected in Hank’s based enzyme-free cell dissociation buffer (Gibco) and resuspended in 500 µL of Opti-MEM. The samples were sonicated as described and then plated onto fully confluent Vero cells that had been washed twice with PBS. A ten-fold serial dilution was performed, and the cells were incubated with virus at 37 °C for 2 h. After 2 h, the Opti-MEM was aspirated, and the DMEM with 0.5% methylcellulose was added. The plates were once again incubated at 37 °C for 72 h before fixation with methanol and staining with crystal violet. The plaques were counted by hand and multiplied by the appropriate dilution factor.

### 2.10. Quantitative PCR Assay

RNA was isolated from HCE cells treated with mock PBS or different concentrations of PMNPs using TRIzol (Life Technologies, Carlsbad, CA, USA) according to the manufacturer’s described protocol. Thus, obtained RNA was quantified using NanoDrop (Thermo Fisher Scientific, Waltham, MA, USA) and equilibrated for all samples with molecular biology grade water (Corning, NY, USA) before they were reverse-transcribed into complementary DNA (cDNA) using the High-Capacity cDNA Reverse Transcription Kit (Applied Biosystems, Foster City, CA, USA). Equal amounts of cDNA were analyzed via real-time quantitative PCR using Fast SYBR Green Master Mix on QuantStudio 7 Flex system (Applied Biosystems). All the primers used in this manuscript were pre-designed and pre-validated from Millipore Sigma Aldrich (KiCqStart^®^ SYBR^®^ Green Primers).

## 3. Results

### 3.1. Preparation of Plasma Membrane Derived Neutralizing Particles (PMNP)

PMNPs were prepared using the two different techniques that involve cellular fractionation. As both the techniques yielded similar results, most of the data shown in this manuscript are based on the PMNPs generated through a column-based kit provided by Invent Biotech (Figure 1).

### 3.2. Characterization of Plasma Membrane Derive Nanoparticles (PMNP)

Sonicating a dispersion of phospholipids in solution creates liposome-like structures. Thus, we subjected the isolated plasma membrane from Vero cells (by procedure mentioned in Figure 1) to sonication in Opti-MEM solution expecting them to form the plasma membrane-derived liposomes. To validate the formation of liposome-like structures, sonicated PMNPs were characterized using bright field microscopy and TEM. A bright field image taken for the PMNPs showed oval shaped nanoparticles less than 1 µm in size (Figure 2A). The TEM results confirmed the isolation of hollow membrane-like particles ranging in size from 850 nm to 200 nm (Figure 2A). A western blot analysis of the samples showed PMNPs isolated from Vero cells consisted of the Na-K ATPase and nectin-1 proteins, which are hallmarks of the plasma membrane fraction (Figure 2B) [22]. The plasma membrane samples derived from CHO cells did not show the expression of nectin 1 as expected. As the PMNPs were isolated from a cellular stock, to standardize the experiments we used a bicinchoninic acid (BCA) protein assay kit (Thermo Fisher Scientific) to quantitate the amount of PMNPs being used for our virus entry and neutralization experiments. We were able to generate a stock solution of 182 µg/mL PMNPs, which was diluted to 100 µg/mL and stored as aliquots. To further assess the cytotoxicity of PMNPs, we performed a MTT assay. HCEs seeded in 96-well plates were subjected to increasing concentrations of PMNPs. Based on the MTT data, cells were viable at all concentrations of PMNPs. Thus, it is safe to say that PMNPs have almost no cytotoxic effects (Figure 2C).

### 3.3. PMNPs Block Viral Entry into Target Cells

Once the PMNPs were isolated, their ability to block viral entry was evaluated. A β-galactosidase-producing reporter virus gL86 was mixed with different concentrations of PMNPs isolated from CHO, HCE or Vero cells for 30 min prior to addition to HCEs (Figure 3A). The results from this experiment showed that PMNPs isolated from HCEs, and Vero cells were able to protect cells from being infected by HSV-1 gL86, while CHO-PMNPs at the highest concentration only showed a 20% reduction in viral entry (Figure 3A). HCE-PMNPs were slightly less efficient than Vero-PMNPs; however, the differences were not significant. In a similar experiment using an RFP reporter HSV-1 virus, we were able to observe that Vero-PMNPs were able to significantly stop HSV-1 from entering the cells when evaluated by imaging (Figure 3B) and by flow cytometry (Figure 3C).

### 3.4. Therapeutic Treatment with PMNPs Protects Human Corneal Epithelial Cells from Viral Spread

After assessing the viral entry inhibition by PMNPs, it was important to evaluate if the PMNPs can deter viral spread. To evaluate this, HCEs were infected with a GFP reporter HSV-1 virus for 2 h followed by the addition of PMNPs (manual isolation) to the cells therapeutically at 100 µg/mL (Figure 4A). The cells were incubated for an additional 22 h prior to analysis. Imaging revealed that HCEs therapeutically treated with HCE-PMNPs and Vero-PMNPs had clear protection in terms of the number of cells infected at the end of 24 h. GFP fluorescence was found in approximately 10% of cells as opposed to 80–90% infected cells in CHO-PMNP-treated and non-treated cells (Figure 4B). A plaque assay conducted using the cell lysates taken from the above-mentioned experiment revealed a greater than 2log10-fold reduction in viral load in HCE-PMNP- and Vero-PMNP- treated cells when compared to mock cells (Figure 4C). A western blot analysis revealed a similar result with a lower amount of HSV-1 gD expression in cells treated with HCE-PMNPs and Vero-PMNPs (Figure 4D). It is important to note that CHO-PMNPs also showed a substantial decrease in viral protein gD; however, the effect was not as pronounced as HCE/Vero-PMNPs.

### 3.5. PMNPs Are Effective against Other Herpesviruses

To further assess whether the PMNPs were active against other herpesviruses, pseudorabies virus (PRV) and herpes simplex virus type-2 (HSV-2) were incubated with different concentrations of Vero-PMNPs (column-based kit) for 30 min and added onto Vero cells. Given that approximately 500 viruses were incubated with varying concentrations of PMNPs, we hypothesized that upon the addition of 5% methylcellulose media to infected Vero-cell monolayers, we would see a reduction in the plaque counts of treated cells when compared to non-treated controls. The results indicate a concentration-dependent reduction in plaque numbers for both the viruses (Figure 5A). Analysis of the plaque counts confirmed our initial findings from the HSV-1 experiments: that there is a significant difference between non-treated and Vero-PMNP-treated groups (Figure 5B).

### 3.6. PMNPs Stimulate Interferon Response

We hypothesized that host cells might identify PMNPs as foreign agents and stimulate an innate immune response leading to the enhanced antiviral effect. To evaluate the potential effect of PMNPs, we treated HCE cells with mock PBS or different concentrations of PMNPs and probed for interferon and other inflammatory cytokine responses associated with initial innate response against foreign particles using quantitative PCR. HCEs were incubated with different concentrations of Vero-PMNPs for a period of 6 h, followed by isolation of RNA to evaluate transcript levels for interferon-α (IFN-A), interferon-β (IFN-B), interleukin-6 (IL6), interleukin-8 (IL8), interleukin-1β (IL-1B), tumor necrosis factor-α (TNF-A) and matrix metalloprotease-9 (MMP9). All of these pro-inflammatory cytokines, if stimulated prior to or during infection, can cause a decrease in viral infectivity. We observed that Vero-PMNPs at a concentration of 100 µg/mL showed a slightly higher but not significant increase in all the aforementioned inflammatory markers (Figure 6). However, at other concentrations of PMNPs, we did not observe any appreciable number of changes.

## 4. Discussion

Cell membrane-derived nanoparticles have been extensively studied in the past, especially as biomimetic, cancer-targeting payloads [20]. The cell membranes act as camouflaging agents for drug payloads that can be delivered to sites of requirement with low immune surveillance. Cell membranes derived from red blood cells, various immune cells, stem cells or cancer cells are extracted and extruded to form vesicles, which can then be used to coat a variety of payloads such as polymers, contrasting agents, cancer drugs, etc. [7,8,9,10,11,12,13,14,15,23,24,25,26,27,28,29,30]. However, in all of the applications mentioned above, the cell membranes are mainly used as carriers of active compounds, and the membrane itself has only been exploited for its biomimetic capability. In this study, we have tried to exploit the ability of membrane-tagged proteins to act as virus neutralizing compounds with minimal modulation of the plasma membrane itself. Plasma membrane-derived liposomes have previously shown efficacy in inhibiting SARS-CoV-2 from infecting cells [31]. In that paper, Zhang et al. suspected that their nanosponges, analogous to our PMNPs, were potentially agnostic to viral mutations and viral species, suggesting a potential application to research in other virology fields. Our research in applying this technology to HSV-1, PRV and HSV-2 confirms this suspicion. Liposomes derived from our Vero and human corneal epithelial (HCE) cell lines demonstrated antiviral activity following incubation with HSV-1. The negative control, CHO-derived PMNPs, demonstrated no such activity. Our initial idea was to isolate plasma membranes from multiple different cell lines and test their efficacy. In this regard, we isolated plasma membranes from Vero, HCE and CHO cells. However, we were surprised to note that beyond a small inhibitory effect at the highest concentration, CHO cell-derived plasma membrane did not have any potent antiviral activity, which was evident in plasma membranes from other two cell types. We hypothesize that this defect in virus neutralization can be attributed to the absence of the cell surface receptor nectin on CHO cells. The small effect seen in CHO cells may be attributed to non-specific ionic interactions between the PMNPs and the virus resulting in neutralizing ability. These cell surface receptors act as decoy receptors for the virus particles, inhibiting the interaction of viral glycoproteins with host cell entry receptors limiting HSV-1 entry into cells. The inflammatory cytokine response and interferon response generated by PMNPs can further aid in the antiviral response shown by PMNPs (Figure 7).

Our results with the PMNPs differ from the nanosponge study in regard to the inclusion of a therapeutic assay against HSV-1. Previously, only a neutralization assay, which entails incubating viruses with the nanosponges prior to their addition to Vero cells, was performed. In this study, we have shown that the PMNPs are not only effective when added concurrently with viruses, but also can reduce viral spread when added therapeutically. Furthermore, the nanosponge study used multiple post-processing steps including the addition of poly (D, L-Lactic-co-glycolic acid) or PLGA cores to the nanosponges. While the addition of polymeric cores ensures consistent particle size across the spectrum, they also increase the number of steps required for the preparation of PMNPs. In this study, we were able to skip the preparation of the polymeric core and still see potent antiviral activity. Although the previous study on nanosponges and our current study on PMNPs show the antiviral benefit associated with these structures, their use as antiviral treatments and drug delivery agents might be of limited interest, given the availability of excellent antivirals and well-characterized delivery agents. Another aspect that remains poorly understood in this manuscript is the directional conformation of the proteins present on the plasma membrane. Future studies will study the conformation of the PMNPs to understand the directionality (inside-out or right-side-out) of the proteins present on them. These results may pave the way for other studies that focus on improving our methodology to achieve greater virus trapping ability and to study viral entry mechanism. Rather than target the replicative process, these liposomes closely resemble their derivatives: human host cell membranes. This resemblance can work to preclude HSV-1 virions from host cell entry since the viruses recognize the PMNPs as viable hosts.

## 5. Conclusions

In conclusion, PMNPs derived from human corneal epithelial cells and Vero cells are effective antiviral agents against HSV-1 infection. PMNPs are effective when used in tandem with virus infection and also when dosed therapeutically. However, their use as antiviral agents of transporters of antiviral drugs might be of limited interested given the availability of various other better characterized agents. The utility of these PMNPs for treatment of HSV-1 needs further optimization of their formulation as well as validation in animal models. However, these PMNPs may be used to better study viral attachment and entry into cells, purely at the plasma membrane, without the interference of other cellular machinery. Our initial report of this new technology might pave the path for many future studies in the virology research field.

## Figures and Tables

**Figure 1 viruses-14-00799-f001:**
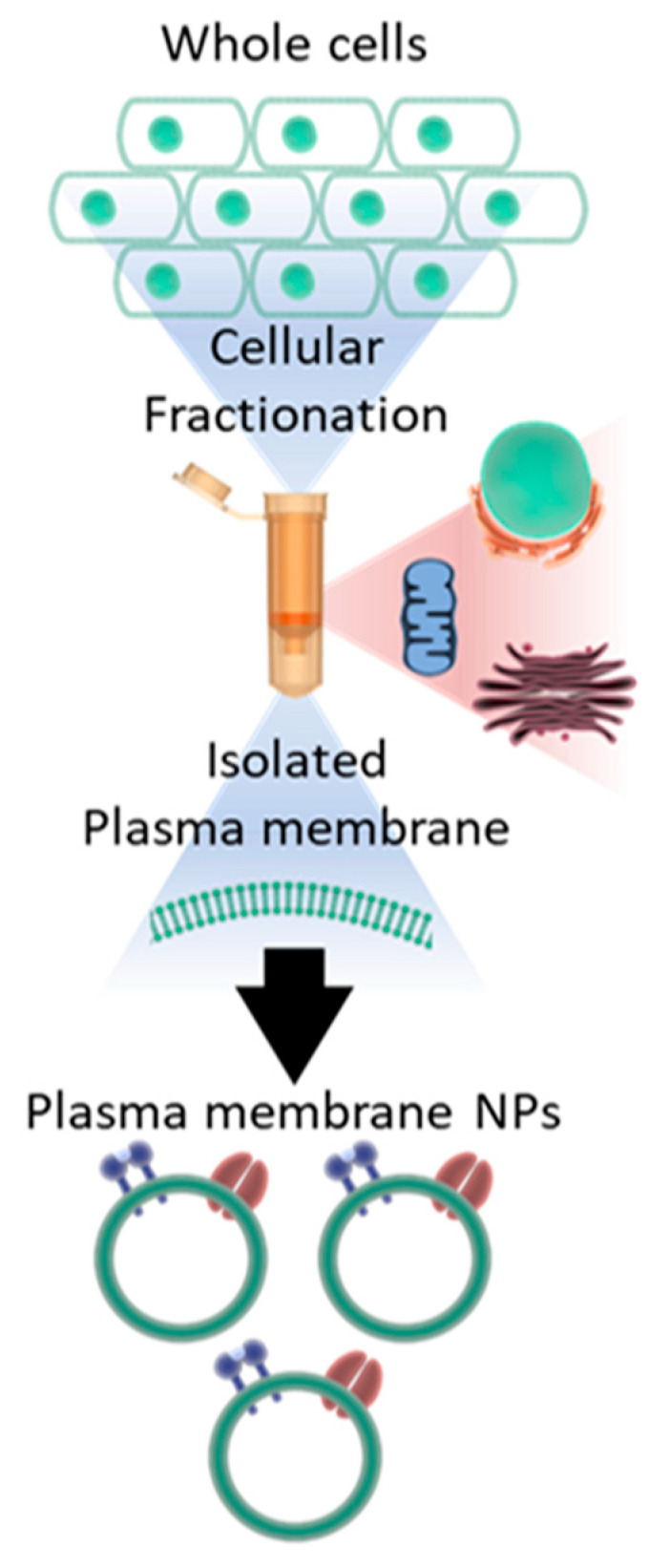
Schematic representation of how the whole cells were fractionated into plasma membrane NPs.

**Figure 2 viruses-14-00799-f002:**
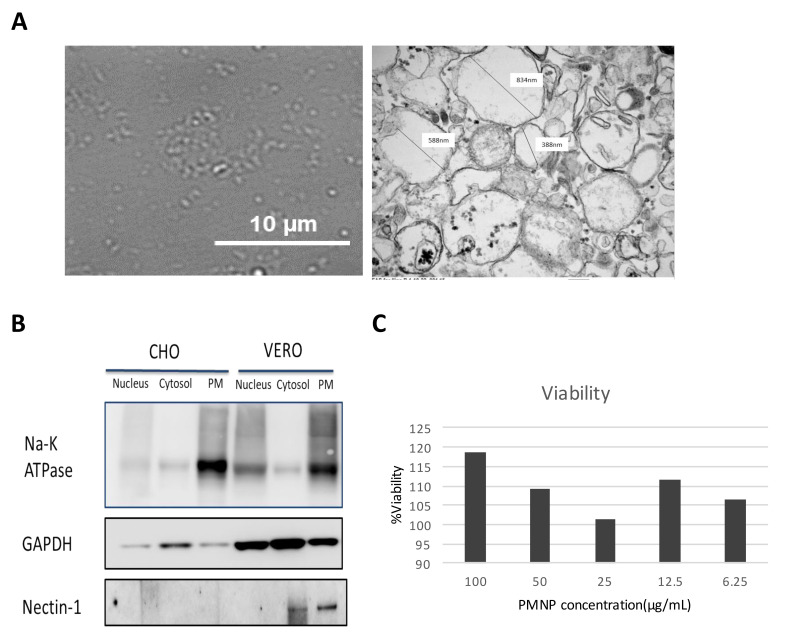
Characterization and validation of the plasma membrane fraction isolated by a column-based kit: (**A**) A 100× image of isolated plasma membrane NPs isolated from Vero cells taken on the brightfield channel of a light microscope. (right) TEM images of plasma membrane particles isolated from Vero cells. (**B**) Western blotting analysis of the various fractions isolated from the whole cell lysate. (**C**) Cell viability of plasma membrane-treated HCE cells assessed by MTT assay.

**Figure 3 viruses-14-00799-f003:**
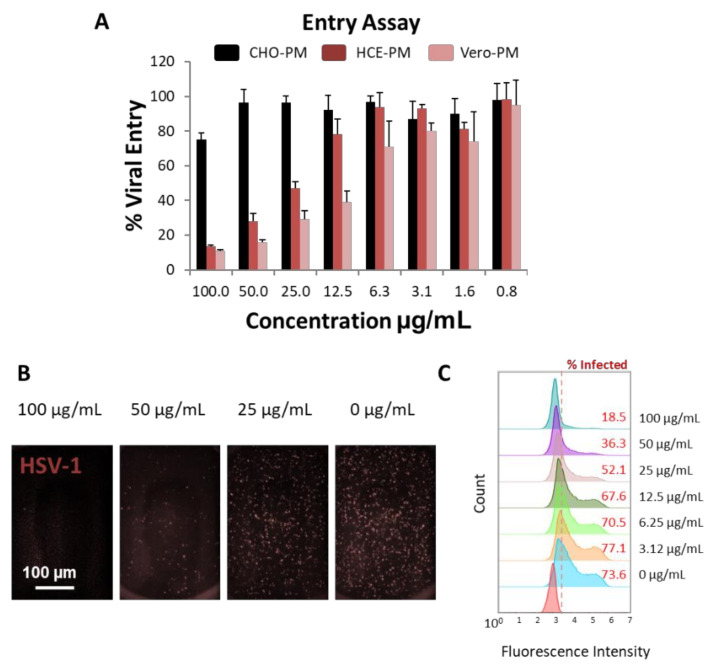
Evaluation of the virus-neutralization ability of PMNPs (isolated by manual isolation technique): (**A**) Results from the β-galactosidase-producing gL86 reporter-based assay to determine viral entry. (**B**) Representative fluorescent images showing the extent of red-fluorescent protein-producing HSV-1 reporter virus present in Vero-PMNP neutralized samples. (**C**) Flow cytometry histograms showing a decrease in the fluorescent population (infected) when virus was neutralized with an increasing concentration of Vero-PMNPs.

**Figure 4 viruses-14-00799-f004:**
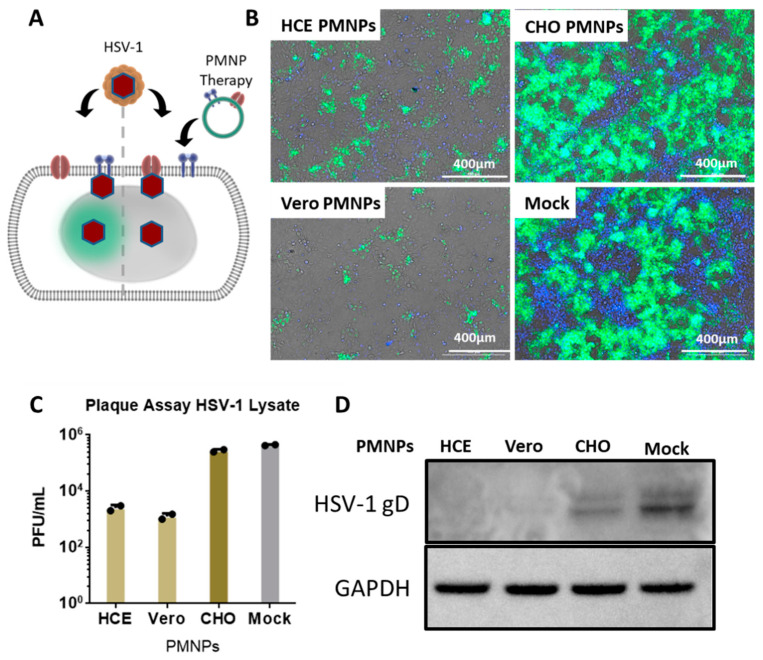
Effect of PMNP therapy of HSV-1 infection: (**A**) A schematic of the therapeutic treatment strategy employed for the experiments. Briefly, HCEs were infected with a GFP-producing reporter virus for 2 h. The media was then replaced with fresh media containing Vero or HCE-PMNPs at a concentration of 100 µg/mL. The cells were incubated for a further 24 h, before (**B**) representative fluorescent images were captured. (**C**) Cell lysates were collected to perform a plaque assay, and the results indicate a reduction in infectious viral titer in samples therapeutically treated with HCE and Vero-PMNPs but not CHO-PMNPs. (**D**) Cell lysates were also used to perform a representative western blot to evaluate viral protein (HSV-1 glycoprotein D) synthesis.

**Figure 5 viruses-14-00799-f005:**
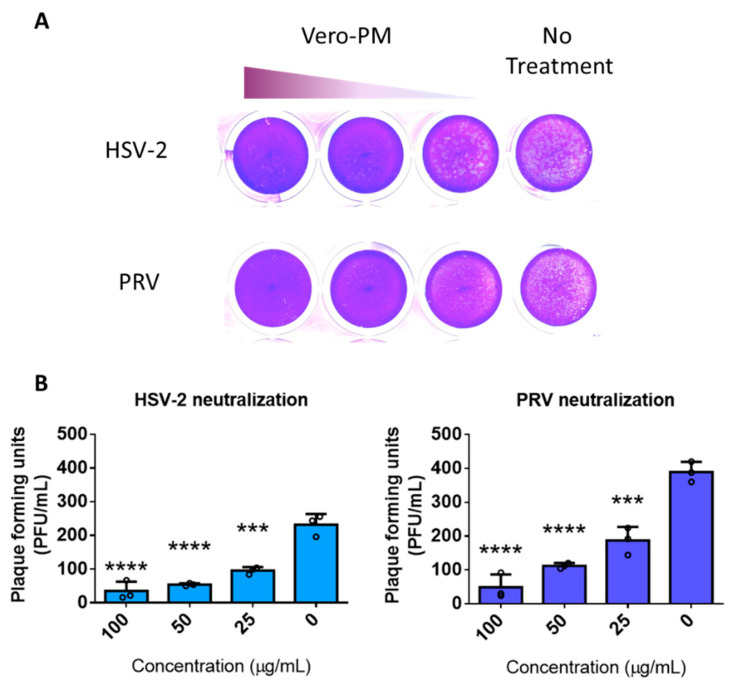
Vero-PMNPs are effective in neutralizing other herpesviruses: (**A**) A plaque reduction assay showing a concentration-dependent reduction in plaques for samples neutralized with Vero-PMNPs. (**B**) Quantification of viral plaques for both HSV-2 and PRV. Experiments were performed in triplicates. A non-parametric, one-way ANOVA analysis was performed comparing to a non-treated sample. *** *p* < 0.001, **** *p* < 0.0001.

**Figure 6 viruses-14-00799-f006:**
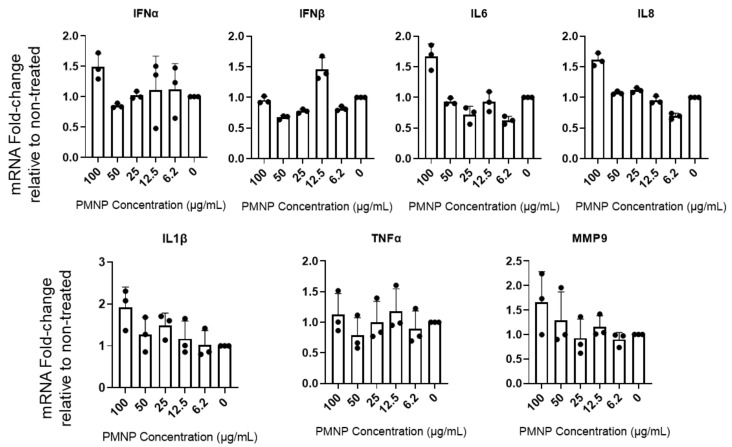
Vero-PMNPs stimulate interferon response. HCE cells were incubated with mock PBS (negative control) or different concentrations of PMNPs. At 24 h post incubation, the cDNA was analyzed for the presence of the desired transcripts using a qRT-PCR machine.

**Figure 7 viruses-14-00799-f007:**
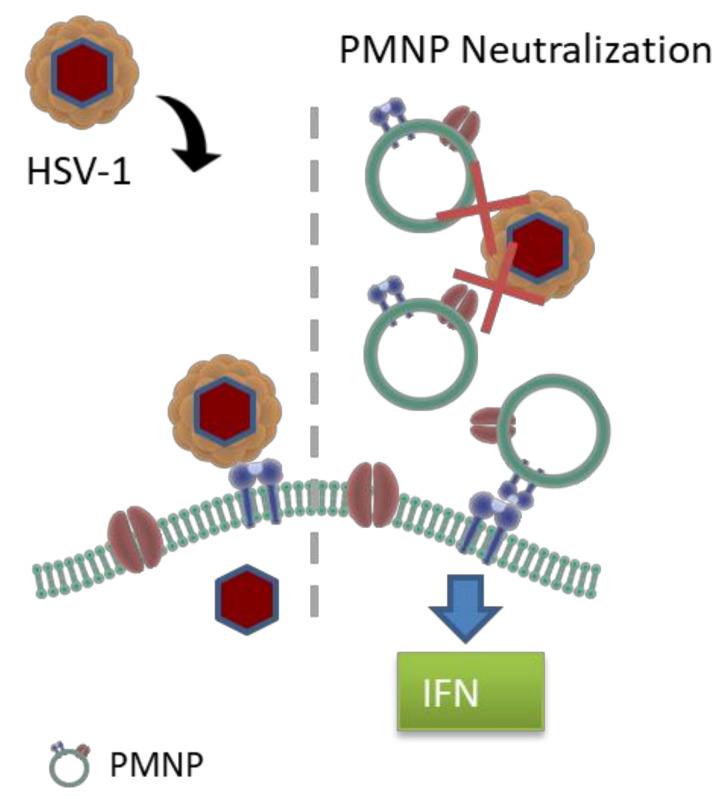
Mechanism of action of virus neutralization by PMNPs. PMNPs acts as decoy to prevent virus binding to host cells. PMNPs at high concentration can also generate an antiviral response by interferon and inflammatory cytokine production.

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
