# Peer review of "Plasma Membrane-Derived Liposomes Exhibit Robust Antiviral Activity against HSV-1"

_viruses, 2022, doi:10.3390/v14040799_

Round 1

Reviewer 1 Report

I am happy that my comments and questions have all been addressed in the revised manuscript.

Thank You

Author Response

Thank you for your review of our manuscript

Reviewer 2 Report

This paper from a group with long time expertise in alphaherpesvirus entry makes some interesting observations. The overall concept of using a receptor decoy to limit viral entry is interesting.  There are two main problems. First, the paper is sloppily composed. Almost none of the Figure calls in the paper correspond to the actual figures. There are a lot of missing methods and undescribed reagents. This is all fixable.  Second, while the authors show a viral inhibition phenomenon, there is nothing at all really about mechanism.

Major:

The authors speculate on but don’t really prove  a mechanism of action for their PNMP preps.  They do show that CHO PMNPs don’t neutralize HSV-1 and they find low nectin-1 in their CHO preps.  HSV-1 can efficiently infect CHO cells. Do the authors know if the mAb used even recognized Chinese hamster nectin-1? There seems to be some literature implying CHO cells don’t express nectin 1 (e.g. Virol J 2006 Dec 27;3:105. doi: 10.1186/1743-422X-3-105) and this should be discussed.   Is it known how HSV-1 gets into CHO cells? A Science article in 1990 implicated an FGF receptor.  Do CHO express HVEM?  Basically, why don’t CHO PMNPs work? 

More basically, why and how do Vero and NCE PMNP’s work? The authors could address the hypothesis that it is decoy binding by nectin 1 by deleting/silencing nectin 1 from these cells with CRISPR, RNAi, etc.  The authors do not address the simple counterhypothesis that Vero and NCE cell PMNPs induce an innate interferon-like response when added to the indicator cells, which then has an antiviral effect. 

The senior author is an expert on HSV entry, especially heparin/heparan proteoglycans.  What is the content of these chemical entities in the PMNPs?  The paper would benefit throughout from introduction and discussion based on what is known about HSV entry, rather than speculating on pharmaceutical preps of PMNP.

The X axis of Fig. 2E is confusing. The authors spend a lot of energy quantitating the PMNP preps in Fig. 2B so they understand the concentration of protein in the prep in micrograms per ml.  Then, in 2E, the term M is used, which usually stands for molarity in standard biomedical use. We are told in line 178 that this actually stands for the number of cells used to make the PMNP prep. Suggest converting the units on the X axis of 2E to micrograms/ml of protein. Also we are not told what cell type was used to make the PMNP used for 2A, 2B, 2C, or 2E.  Since the authors use multiple cell types to make PMNP in this paper, please be specific.

Fig. 2D what is the interpretation of the nectin 1 immunoblot for the CHO cells?  I don’t see a band.  Are we supposed to?  Does the mAb used react with Chinese hamster (CHO), non human primate (Vero) and human (NCE).

Line 195 mentions an RFP reporter virus for the first time in the paper. Please provide details in the methods section including parent strain and reference.

Fig 3C is uninterpretable. There are supposed to be 8 curves but I can only see about 5 and the differences in the pink shades and lines are subtle.  Can the authors make the figure clearer or simplify to a percent positive cells at various concentrations of PMNP?

The anti gD mab used in 4D needs to be specified. 4B micrograph needs a scale bar.

The line 296 material about loading PMNPs with drugs is very speculative and should be deleted. Liposomal drug delivery with chemically defined liposomes is used for a licensed antifungal drug and at least one chemotherapeutic drug and is greatly preferred in the pharma industry to an undefined natural product.

Minor:

Can we learn anything more about the HCE cells?  Are they primary cells? Are they immortalized in any way?  Do they need to be used at a certain passage number before they change characteristics?

Line 76 do the authors mean proton pump inhibitor or protease inhibitor? 

P90 the details of sonication may matter for others to reproduce.  Please provide some details (probe, bath, manufacturer, settings, times, temps, etc.).

Line 59 mentions HSV-1 17 but line 93/127 mention HSV-1 (17GFP) which sounds like a genetically modified derivative. Please explain/reconcile. Do the authors actually mean RFP (line 195)?

Line 94 temp for incubation?

Line 101 please provide citation and basic information about HSV-1 gL86.  What is the parent strain?  This seems to have beta gal in it.  Please give reference to creation.

Line 108 what does PMNP mean? (line 154 explains but explain at first use)

The standard curve in 2B is not necessary and in any case the actual data points are not shown; all we see are superimposed blue and black lines which don’t mean a lot.

Line 165-180 region the text talks about Fig 1 but actually the mean Fig 2.  Correct this.  There is not text call to Fig 2E.  Figure calls in 189-198 region are also all wrong.  Figure calls in line 209-221 region are all wrong.  Double check and make sure figure calls correspond to figures.

Line 306-313 is the journal template.

Author Response

REPLY TO REVIEWER COMMENTS:

Comments from the reviewer

This paper from a group with long time expertise in alphaherpesvirus entry makes some interesting observations. The overall concept of using a receptor decoy to limit viral entry is interesting.  There are two main problems. First, the paper is sloppily composed. Almost none of the Figure calls in the paper correspond to the actual figures. There are a lot of missing methods and undescribed reagents. This is all fixable.  Second, while the authors show a viral inhibition phenomenon, there is nothing at all really about mechanism.

Major:

The authors speculate on but don’t really prove  a mechanism of action for their PNMP preps.  They do show that CHO PMNPs don’t neutralize HSV-1 and they find low nectin-1 in their CHO preps.  HSV-1 can efficiently infect CHO cells. Do the authors know if the mAb used even recognized Chinese hamster nectin-1? There seems to be some literature implying CHO cells don’t express nectin 1 (e.g. Virol J 2006 Dec 27;3:105. doi: 10.1186/1743-422X-3-105) and this should be discussed.   Is it known how HSV-1 gets into CHO cells? A Science article in 1990 implicated an FGF receptor.  Do CHO express HVEM?  Basically, why don’t CHO PMNPs work?

More basically, why and how do Vero and NCE PMNP’s work? The authors could address the hypothesis that it is decoy binding by nectin 1 by deleting/silencing nectin 1 from these cells with CRISPR, RNAi, etc.  The authors do not address the simple counterhypothesis that Vero and NCE cell PMNPs induce an innate interferon-like response when added to the indicator cells, which then has an antiviral effect.

CHOs are non-permissive to HSV-1 infection [PMIDs: 8898196; 9657005; 9616127]. They don’t express nectin or HVEM. And that is the reason we hypothesized that it does not act as receptor decoy to limit viral entry. We have now added this reference to support our claim in the manuscript along with a brief explanation. While the article referred by the reviewer has implicated fgf receptor as an entry receptor for HSV into CHO cells, this study has not been corroborated by other experts and refuted in a response published in Science (PMID: 1649495). Multiple studies since then have shown the non-permissive nature of CHO to HSV infection.

We thank reviewer for the suggestion relating to gene silencing to decipher our results better. However, this is currently beyond the scope of the current manuscript and will be taken into consideration for our future study.

Based on the comments provided by the reviewer, we have performed the requisite experiments and added a section to address the counter hypothesis section in Figure 6 about cytokine and interferon responses.

The senior author is an expert on HSV entry, especially heparin/heparan proteoglycans.  What is the content of these chemical entities in the PMNPs?  The paper would benefit throughout from introduction and discussion based on what is known about HSV entry, rather than speculating on pharmaceutical preps of PMNP.

The contents of Vero cell plasma membrane have been elucidated in the past (PMC3880917). The cited article goes into much depth regarding how viruses use these proteins for viral entry. This article has now been cited in our manuscript to give the reader a better perspective of what proteins may make up the PMNPs.

The X axis of Fig. 2E is confusing. The authors spend a lot of energy quantitating the PMNP preps in Fig. 2B so they understand the concentration of protein in the prep in micrograms per ml.  Then, in 2E, the term M is used, which usually stands for molarity in standard biomedical use. We are told in line 178 that this actually stands for the number of cells used to make the PMNP prep. Suggest converting the units on the X axis of 2E to micrograms/ml of protein. Also we are not told what cell type was used to make the PMNP used for 2A, 2B, 2C, or 2E.  Since the authors use multiple cell types to make PMNP in this paper, please be specific.

We apologize for the confusion. We have made the changes to the concentrations to make it more lucid. We have also added the cell type used to create PMNPs in that figure.

Fig. 2D what is the interpretation of the nectin 1 immunoblot for the CHO cells?  I don’t see a band.  Are we supposed to?  Does the mAb used react with Chinese hamster (CHO), non human primate (Vero) and human (NCE).

CHO cells do not express nectin-1 [PMID: 9616127]. The only time nectin-1 can be seen is by transfecting these cells with nectin-1 expression plasmid [PMID: 9616127], therefore, we are not supposed to see a band. The nectin-1 antibody purchased does not specifically refer for its use in Monkey or Hamster cells. However, we have used this antibody in the past to detect Nectin-1 in Vero cells.

Line 195 mentions an RFP reporter virus for the first time in the paper. Please provide details in the methods section including parent strain and reference.

Thanks for addressing this. We have added the RFP virus detail in the paper.

Fig 3C is uninterpretable. There are supposed to be 8 curves but I can only see about 5 and the differences in the pink shades and lines are subtle.  Can the authors make the figure clearer or simplify to a percent positive cells at various concentrations of PMNP?

We have re-plotted figure 3c using FlowJo software to ensure all the curves are better visualized.

The anti gD mab used in 4D needs to be specified. 4B micrograph needs a scale bar.

We have added the gD mAb used in the methods section. The original 4B had a scale bar which was not clearly visible. We have corrected that for 4B.

The line 296 material about loading PMNPs with drugs is very speculative and should be deleted. Liposomal drug delivery with chemically defined liposomes is used for a licensed antifungal drug and at least one chemotherapeutic drug and is greatly preferred in the pharma industry to an undefined natural product.

We have deleted the line.

Minor:

Can we learn anything more about the HCE cells?  Are they primary cells? Are they immortalized in any way?  Do they need to be used at a certain passage number before they change characteristics?

We have mentioned in our materials and methods that we have HCEs provided by Kozaburo Hayashi (National Eye Institute). We have added the reference for additional details about the cell line.

Line 76 do the authors mean proton pump inhibitor or protease inhibitor?

Thanks for pointing this out. We have corrected it to protease and phosphatase inhibitor.

P90 the details of sonication may matter for others to reproduce.  Please provide some details (probe, bath, manufacturer, settings, times, temps, etc.).

We have added the exact sonication details used by us.

Line 59 mentions HSV-1 17 but line 93/127 mention HSV-1 (17GFP) which sounds like a genetically modified derivative. Please explain/reconcile. Do the authors actually mean RFP (line 195)?

We have used HSV-1 17syn+ parent strain for most of the experiments. We have used the HSV-1 17gfp strain for Figure 4b only. We have added the citation for both strains in the methods section.

Line 94 temp for incubation?

We have added the incubation temperature to the requisite section.

Line 101 please provide citation and basic information about HSV-1 gL86.  What is the parent strain?  This seems to have beta gal in it.  Please give reference to creation.

HSV-1 gL86 was made from KOS (Kendall O’Smith) parent strain and has been used extensively in the past by our group and others. We have added the citation for the strain.

Line 108 what does PMNP mean? (line 154 explains but explain at first use)

Thank you for pointing this out. We have now explained PMNP at the first instance.

The standard curve in 2B is not necessary and in any case the actual data points are not shown; all we see are superimposed blue and black lines which don’t mean a lot.

We have removed the 2B data figure.

Line 165-180 region the text talks about Fig 1 but actually the mean Fig 2.  Correct this.  There is not text call to Fig 2E.  Figure calls in 189-198 region are also all wrong.  Figure calls in line 209-221 region are all wrong.  Double check and make sure figure calls correspond to figures.

Line 306-313 is the journal template.

We apologize those figure call issues. We have checked and made those changes.

Round 2

Reviewer 2 Report

Critiques have been met and paper is acceptable in present form.